# The DEPTQ^+^ Experiment: Leveling the DEPT Signal Intensities and Clean Spectral Editing for Determining CH_n_ Multiplicities

**DOI:** 10.3390/molecules26123490

**Published:** 2021-06-08

**Authors:** Peter Bigler, Camilo Melendez, Julien Furrer

**Affiliations:** Departement für Chemie, Biochemie und Pharmazie, Universität Bern, Freiestrasse 3, CH-3012 Bern, Switzerland; camilo.melendez@dcb.unibe.ch

**Keywords:** NMR, ^13^C, DEPTQ, DEPTQ^+^, spectral editing, *J*-cross talk, signal intensity leveling

## Abstract

We propose a new ^13^C DEPTQ^+^ NMR experiment, based on the improved DEPTQ experiment, which is designed to unequivocally identify all carbon multiplicities (Cq, CH, CH_2_, and CH_3_) in two experiments. Compared to this improved DEPTQ experiment, the DEPTQ^+^ is shorter and the different evolution delays are designed as spin echoes, which can be tuned to different ^1^*J*_CH_ values; this is especially valuable when a large range of ^1^J_CH_ coupling constants is to be expected. These modifications allow (i) a mutual leveling of the DEPT signal intensities, (ii) a reduction in *J* cross-talk in the Cq/CH spectrum, and (iii) more consistent and cleaner CH_2_/CH_3_ edited spectra. The new DEPTQ^+^ is expected to be attractive for fast ^13^C analysis of small-to medium sized molecules, especially in high-throughput laboratories. With concentrated samples and/or by exploiting the high sensitivity of cryogenically cooled ^13^C NMR probeheads, the efficacy of such investigations may be improved, as it is possible to unequivocally identify all carbon multiplicities, with only one scan, for each of the two independent DEPTQ^+^ experiments and without loss of quality.

## 1. Introduction

Two-dimensional heteronuclear NMR techniques for ^13^C assignment have long been considered extremely powerful since they are much more sensitive than, e.g., one-dimensional (1D) ^13^C spectral editing experiments [1], and the corresponding spectra contain a lot of information. Nevertheless, “old” 1D experiments such as SEFT [2], APT [3] and PENDANT [4,5], refocused variants of INEPT [6], DEPT [7,8], DEPTQ [9,10,11] and SEMUT [12], continue to be very useful for routine applications [13]. A likely reason for the popularity of these 1D experiments is the rather low resolution achieved in the indirect dimension (^13^C) with 2D heteronuclear experiments, such as HSQC [14], HMBC [15,16,17] or LR-HSQMBC [18], under standard experimental conditions, thus potentially preventing the unambiguous differentiation between closely spaced ^13^C resonances. In principle, this problem can easily be alleviated [19,20] but the required experiments/methods can only be routinely applied to a certain extent, as they require in-depth knowledge of theoretical and experimental NMR spectroscopy.

One-dimensional ^13^C experiments such as APT [2], PENDANT [4,5], and DEPTQ [9,10,11], provide information about both quaternary and protonated carbons in the spectra, with the signals of Cq and CH_2_ carbons being 180° out-of-phase with respect to the signals of CH and CH_3_ carbons. Among those experiments, the DEPTQ encompasses all the known advantages of the basic DEPT experiment but includes the signals of quaternary carbons. The DEPT pulse sequences, thanks to their marked insensitivity to experimental parameters compared to other experimental schemes, have proven to be the experiments of choice for obtaining the chemical shift and multiplicity information for all types of protonated carbons [8]. Additionally, a version of DEPTQ with the length of the initial ^1^H pulse adjusted accordingly allows simplified spectral editing: with this first and the final proton pulse adjusted to 45° and 90°, respectively, for a DEPTQ90 (DEPTQ90_45_) and adjusted to 90° and 135°, respectively, for a DEPTQ135 (DEPTQ135_90_), the intensities of CH signals are equal in both experiments, since they are reduced to 70% of their maximum value with the DEPTQ90_45_ experiment. Thus, subtracting the DEPTQ90_45_ and the DEPTQ135_90_ from each other yields a CH_2_/CH_3_ edited spectrum with the signals of the two carbon types in opposite phase [11]. The DEPTQ90 spectrum shows exclusively, and with opposite phase, the signals of Cq and CH.

Therefore, although no Cq-. CH-, CH_2_- and CH_3_-only spectra result, the pairwise editing with opposite signs for the signals of different multiplicities allows unambiguous spectral editing with only two experiments, and conceivably, with only one scan each, e.g., for highly concentrated samples [11]. 

Experimentally, we found that with the DEPTQ experiment, the editing filter quality in the Cq/CH and CH_2_/CH_3_ subspectra is not satisfactory, with a breakthrough of many artifacts originating from other CH_n_ multiplicities [13]. We also observed that for molecules possessing large ranges of ^1^*J*_CH_ coupling constants, DEPTQ135 spectra and edited subspectra are obtained, with missing or very weak signals, especially CH_3_ or alkyne CH groups (Figure 1). For instance, in the DEPTQ135_90_ spectrum of 4-methyl-*N,N*-di(prop-2-yn-1-yl)aniline, the resonance of the methyl group C8 at 20.6 ppm is almost invisible (Figure 1). While the DEPTQ90_45_ spectrum appears satisfactory, the edited CH_2_/CH_3_ subspectrum is misleading, as the resonance of the methyl group C8 at 20.6 ppm is absent. We thought it useful to develop a modified DEPTQ scheme for obtaining consistent DEPTQ and edited subspectra for all classes of molecules. This could potentially be very attractive for rapid ^13^C analysis of newly synthesized or isolated compounds, especially for high-throughput NMR analysis. 

In this article, we present a new pulse sequence, DEPTQ^+^, possessing all these attributes, and compare it with the modified DEPTQ experiment, designed to unequivocally identify all carbon multiplicities (Cq, CH, CH_2_, and CH_3_) in two experiments. We show both theoretically and experimentally that the DEPTQ^+^ pulse sequence always provides unambiguous DEPTQ spectra with a leveling of all carbon intensities. This generally results in processed spectra with a clean separation of the Cq and CH signals in one spectrum and the CH_2_ and CH_3_ signals in the other, with both spectra showing the signals of the two carbon species in opposite phases. It must be emphasized, however, that these properties only come into play and are relevant for molecules possessing a large range of ^1^*J*_CH_ coupling constants.

## 2. Results and Discussion

### 2.1. Theory

#### 2.1.1. Original DEPTQ Pulse Sequence

Despite their known tolerance to the experimental parameters, edited DEPTQ spectra may show undesirable residual CH_n_ signals from “wrong” carbon types, a phenomenon termed dubbed *J* cross-talk [21]. These signals arise, among other things, with non-ideally adjusted delays and occur for both the coherences resulting from initial ^1^H- and from ^13^C-polarization. These signals are particularly troublesome in the pairwise edited CH_n_ spectra, as they can lead to misinterpretation, particularly if the editing is performed in an automatic or semi-automatic way and with samples containing several, differently concentrated components [22,23,24,25,26,27]. Therefore, this study predominantly focuses on suppressing remaining CH_n_ signals in the different subspectra as efficiently as possible. 

The general condition δ ≠ 1/(2 * ^1^*J*_CH_) is assumed in the following first product operator treatments of the original DEPTQ pulse sequence (Figure 2). For simplicity, ^1^H- and ^13^C-shift evolutions during the delays δ and the gradients G_1_–G_3_ used in practice are not considered (the overall result is identical, except that only one of the ^13^C magnetization terms at coherence level −1, C^−^, is detected if the PFGs are considered). We use here the bracket notations proposed by Mateescu and Valeriu [28]. 

##### Quaternary Carbons

The first ^13^C pulse φ is set arbitrarily (Ernst angle)
(1)$$\begin{array}{cc} \left[z\right]\overset{\emptyset {}^{\circ}yC}{\to }cos\emptyset \left[z\right] & +sin\emptyset \left[x\right]\overset{\delta }{\to }cos\emptyset \left[z\right]+sin\emptyset \left[x\right]\overset{\Omega {}^{\circ}xH}{\to }\overset{180{}^{\circ}xC}{\to }\overset{\delta }{\to }-cos\emptyset \left[z\right]\\ & +sin\emptyset \left[x\right]\overset{180{}^{\circ}xH}{\to }\overset{90{}^{\circ}xC}{\to }cos\emptyset \left[y\right]+sin\emptyset \left[x\right]\overset{\delta }{\to }cos\emptyset \left[y\right]\\ & +sin\emptyset \left[x\right]\overset{180{}^{\circ}xC}{\to }\overset{\theta {}^{\circ}yH}{\to }-cos\emptyset \left[y\right]+sin\emptyset \left[x\right]\overset{\delta }{\to }\overset{90{}^{\circ}-xC}{\to }cos\emptyset \left[z\right]\\ & +sin\emptyset \left[x\right] \end{array}$$

Of course, the outcome for quaternary carbons is identical in both experiments, DEPTQ135_90_, Ω = 90° and DEPTQ90_45_, Ω = 45°, irrespective of the length of the proton pulse θ.

DEPTQ135, Ω = 90°, θ = 135°

Subsequently, the following abbreviations will be used:c = cos π*J*δ and s = sin π*J*δ  c2 = cos 2*πJ*_CH_δ and s2 = sin 2π*J*_CH_δ

For the observable signals originating from the polarization transfer from ^1^H to ^13^C (DEPT135), we obtain [7,8,29]: 

1. CH groups
(2)$$\begin{array}{l} \left[1z\right]\overset{\emptyset {}^{\circ}yC}{\to }\overset{\delta }{\to }\left[1z\right]\overset{90{}^{\circ}xH}{\to }\overset{180{}^{\circ}xC}{\to }-\left[1y\right]\overset{\delta }{\to }s\left[zx\right]\overset{\left(180{}^{\circ}\right)xH}{\to }\overset{\left(90{}^{\circ}\right)xC}{\to }-s\left[yx\right]\overset{\delta }{\to }\\ -s\left[yx\right]\overset{\left(180{}^{\circ}\right)xC}{\to }s\left[yx\right]\overset{135{}^{\circ}yH}{\to }0.7s\left[yx\right]\\ +0.7s\left[yz\right]\overset{\delta }{\to }\overset{90{}^{\circ}-xC}{\to }\overset{1_{H-decoupling}}{\to }0.7s^{2}\left[x1\right] \end{array}$$

2. CH_2_ groups
(3)$$\left[1z1\right]+\left[11z\right]\overset{\emptyset {}^{\circ}yC}{\to }\overset{\delta }{\to }\ldots \overset{\delta }{\to }\overset{90{}^{\circ}-xC}{\to }0.35s2^{2}\left[x11\right]-s^{4}\left[x11\right]$$

3. CH_3_ groups
(4)$$\begin{array}{l} \left[1z11\right]+\left[11z1\right]+\left[111z\right]\overset{\emptyset {}^{\circ}yC}{\to }\overset{\delta }{\to }\ldots \overset{\delta }{\to }\overset{90{}^{\circ}-xC}{\to }0.53s2^{2}c^{2}\left[x111\right]\\ -0.75s2^{2}s^{2}\left[x111\right]+1.06s^{6}\left[x111\right] \end{array}$$

The final carbon signal intensities, represented by *I*, for the three groups CH, CH_2_ and CH_3_ for Ω = 90° and θ = 135°, as a function of the evolution delay δ and the actual coupling constant *J_CH_*, are thus (the operators [*x*1], [*x*11], and [*x*111] have been replaced by the Cartesian operator *C_x_* for simplicity):
(5)*I_CH_* = *C_x_* [0.71 sin^2^(π*J_CH_*δ)]*I_CH_*_2_ = *C_x_* [0.35 sin^2^(2π*J*_CH_δ) − 1.00 sin^4^(π*J*_CH_δ)]*I_CH_*_3_ = *C_x_* [0.53 sin^2^(2π*J*_CH_δ) cos^2^(π*J*_CH_δ) − 0.75 sin^2^(2π*J*_CH_δ) sin^2^(π*J*_CH_δ) + 1.06sin^6^(π*J*_CH_δ)]

which reduces with δ = 1/(2 * ^1^*J*_CH_) (matched delay) to:
(6)*I_CH_* = +0.71 *C_x_**I_CH_*_2_ = −1.00 *C_x_**I_CH_*_3_ = +1.06 *C_x_*

Equation (5) provides the analytical expressions to calculate the carbon signal intensities for the three groups CH, CH_2_ and CH_3_ as a function of the evolution delay δ and the actual coupling constant *J_CH_.* Obviously, very weak resonances result in cases of large mismatches between actual ^1^*J*_CH_ coupling constants and the average ^1^*J*_CH_^0^ coupling constant set (=1/(2 * δ)) in the DEPTQ experiment. In detail, large mismatches exist (i) for acetylenic CH (^1^*J*_CH_ ~ 250 Hz) or hetero aromatic rings (^1^*J*_CH_ ~ 170–210 Hz) when ^1^*J*_CH_^0^ is set to the classical value of 145 Hz, and (ii) for typical methyl groups (^1^*J*_CH_ ~ 110–135 Hz) when ^1^*J*_CH_^0^ is set to an average value 185 Hz, to account for the whole range of ^1^*J*_CH_ coupling constants (100–250 Hz) when acetylenic CH or hetero aromatic rings are present (Appendix A). 

DEPTQ90, Ω = 45°, θ = 90°

1. CH groups
(7)$$\begin{array}{l} \left[1z\right]\overset{\emptyset {}^{\circ}yC}{\to }\overset{\delta }{\to }\overset{180{}^{\circ}xC}{\to }\left[1z\right]\overset{45{}^{\circ}xH}{\to }-0.7\left[1y\right]\overset{\delta }{\to }0.7s\left[zx\right]\overset{\left(180{}^{\circ}\right)xH}{\to }\overset{\left(90{}^{\circ}\right)xC}{\to }-0.7s\left[yx\right]\overset{\delta }{\to }\\ -0.7s\left[yx\right]\overset{\left(180{}^{\circ}\right)xC}{\to }0.7s\left[yx\right]\overset{90{}^{\circ}yH}{\to }-0.7s\left[yz\right]\overset{\delta }{\to }\overset{90{}^{\circ}xC}{\to }0.7s^{2}\left[x1\right] \end{array}$$

2. CH_2_ groups
(8)$$\left[1z1\right]+\left[11z\right]\overset{\emptyset {}^{\circ}yC}{\to }\overset{\delta }{\to }\ldots \overset{\delta }{\to }\overset{90{}^{\circ}-xC}{\to }0.35s2^{2}\left[x11\right]$$

3. CH_3_ groups
(9)$$\left[1z11\right]+\left[11z1\right]+\left[111z\right]\overset{\emptyset {}^{\circ}yC}{\to }\overset{\delta }{\to }\ldots \overset{\delta }{\to }\overset{90{}^{\circ}-xC}{\to }2.11s^{2}c^{4}\left[x111\right]$$

The final carbon signal intensities, represented by *I*, for the three groups CH, CH_2_, and CH_3_ for Ω = 45° and θ = 90°, as a function of the evolution delay δ and the actual coupling *J_CH_*, are thus:
(10)*I_CH_* = *C_x_* [0.71 sin^2^(π*J_CH_*δ)]*I_CH_*_2_ = *C_x_* [0.35 sin^2^(2π*J*_CH_δ)]*I_CH_*_3_ = *C_x_* [2.11 sin^2^(π*J*_CH_δ) cos^4^(π*J*_CH_δ)]

which reduces with *δ* = 1/(2 * ^1^*J*_CH_) (matched delay) to:
(11)*I_CH_* = +0.71 *C_x_**I_CH_*_2_ = 0*I_CH_*_3_ = 0

DEPTQ135_90_ − DEPTQ90_45_, edited CH_2_/CH_3_ subspectrum

The subtraction of the two data, DEPTQ135_90_ − DEPTQ90_45_, leads to:(12)*I_CH_* = *C_x_* [0.71 sin^2^(π*J_CH_*δ)] − *C_x_* [0.71 sin^2^(π*J_CH_*δ)] = 0*I_CH_*_2_ = *C_x_* [0.35 sin^2^(2π*J*_CH_δ) − 1.00 sin^4^(π*J*_CH_δ)] − *C_x_* [0.35 sin^2^(2π*J*_CH_δ)] = −*C_x_* sin^4^(π*J*_CH_δ)*I_CH_*_3_ = *C_x_* [0.53 sin^2^(2π*J*_CH_δ) cos^2^(π*J*_CH_δ) − 0.75 sin^2^(2π*J*_CH_δ) sin^2^(π*J*_CH_δ) + 1.06 sin^6^(π*J*_CH_δ)] − *C_x_* [2.11 sin^2^(π*J*_CH_δ) cos^4^(π*J*_CH_δ)]

which reduces with *δ* = 1/(2 * ^1^*J*_CH_) (matched delay) to:(13)*I_CH_*_2_ = −1.00 *C_x_**I_CH_*_3_ = +1.06 *C_x_*

For the typical *J*-coupling constant ranges (for which δ is usually set to an average value of 145 Hz), the artefacts originating from CH_2_ groups in the DEPTQ90_45_ spectrum are expected to be weak: due to its sin^2^-dependence, the intensity of the cross-talk term 0.35 sin^2^(2π*J*_CH_δ) in Equation (10) is generally small, which is also shown in Appendix A. However, CH_2_ artifacts can also be very intense and actually troublesome when the full *J*-coupling constant range is taken into account (100–250 Hz), for which δ is usually set to a larger average value around 180 Hz (Appendix A) [19,30]. Likewise, for a CH_3_ group, the intensity of the cross-talk term 2.11 sin^2^(π*J*_CH_δ) cos^4^(π*J*_CH_δ) in Equation (10) and, thus, CH_3_ artifacts, will be very weak for the usual *J*-coupling constant ranges, as shown in Appendix A, but can also be very intense and troublesome when the full *J*-coupling constant range is considered (Appendix A) [19,30]. 

The DEPTQ experiment has two potential drawbacks: (i) the problem of missing or very weak resonances in the DEPTQ135_90_ when large ^1^*J*_CH_ coupling constant ranges are considered, and (ii) the existence of artefacts, possibly strong, originating from CH_2_- and CH_3_ groups in the DEPTQ90_45_ spectrum. We propose the modification of the DEPTQ experience to remedy these two weaknesses.

#### 2.1.2. The DEPTQ^+^ Pulse Sequence

The proposed DEPTQ^+^ pulse sequence is depicted in Figure 2 together with the original DEPTQ pulse sequence [10]. Compared to the latter, the evolution periods of length δ = 1/(2 * ^1^*J*_CH_) are replaced by three spin echoes of different length [21,30]. Interestingly, these spin-echoes make it possible to omit the first evolution period in the standard DEPTQ pulse sequence that must precede the DEPT pulse sequence for proper refocusing of the Cq-magnetization generated with the first ^13^C excitation pulse. Therefore, as in the PENDANT sequence [4,5], both the ^1^H and ^13^C excitation pulses are synchronized. The total length of DEPTQ^+^ is, thus, shorter than that of DEPTQ and identical to a standard DEPT experiment, which will reduce the relaxation losses.

The advantages of the DEPTQ^+^ compared to the DEPTQ experiment, which result from the individual tuning of the three spin echo periods to different ^1^*J*_CH_ coupling constants are: (i) a mutual leveling of the DEPT signal intensities, similar to that achieved with the ACCORD-DEPT experiment, which systematically samples a range of ^1^*J*_CH_ coupling constants [31]. The accordion principle is, however, more complicated to implement than tuning the three spin echo periods, as shown subsequently. (ii) A reduction in *J* cross-talk in the CH/Cq spectrum (DEPTQ90), although this *J* cross-talk reduction is only noticeable and disturbing for molecules with a large range of ^1^*J*_CH_ coupling constants, as shown theoretically (Appendix A), and subsequently in the experimental part. 

Note that the intensities of quaternary carbon resonances are, of course, not affected by these three echo periods used with DEPTQ^+^ and are the same as with DEPTQ. However, compared to the DEPTQ experiment, note that while the first ^13^C pulse φ is ≠90°, the additional 180° pulses during the three spin echo periods invert the sign of the residual *z*-magnetization that is not detected. Thus, the phase of the additional 90° ^13^C pulse prior to data acquisition must be applied along the *+x* axis to re-establish the residual ^13^C-*z* magnetization left with the initial ^13^C pulse.

##### Quaternary Carbons

The first ^13^C pulse *ϕ* is set arbitrarily (Ernst angle):(14)$$\begin{array}{cc} \left[z\right]\overset{\emptyset {}^{\circ}yC}{\to }\overset{\delta_{1}}{\to }\overset{180{}^{\circ}xC}{\to } & -cos\emptyset \left[z\right]+sin\emptyset \left[x\right]\overset{90{}^{\circ}xC}{\to }cos\emptyset \left[y\right]\\ & +sin\emptyset \left[x\right]\overset{\delta_{2}}{\to }\overset{180{}^{\circ}xC}{\to }\overset{\delta_{3}}{\to }\overset{180{}^{\circ}xC}{\to }cos\emptyset \left[y\right]+sin\emptyset \left[x\right]\overset{90{}^{\circ}xC}{\to }cos\emptyset \left[z\right]\\ & +sin\emptyset \left[x\right] \end{array}$$

The product operator evaluation for the CH_n_ coherences originating from initial ^1^H-polarization is applied in the DEPTQ^+^ pulse sequence, using the following abbreviations:

c = cos (π*J*_CH_δ_1_), s = sin (π*J*_CH_δ_1_)

c’ = cos (π*J*_CH_δ_2_), s’ = sin (π*J*_CH_δ_2_)

c’’ = cos (π*J*_CH_δ_3_), s’’ = sin (π*J*_CH_δ_3_)

s2’’ = sin (2π*J*_CH_δ_3_)

DEPTQ^+^135_90_, Ω = 90°, θ = 135°

The final carbon signal intensities, represented by *I*, for the three groups CH, CH_2_, and CH_3_ for Ω = 90°, θ = 135°, as a function of the evolution delay δ, and the actual coupling constant *J_CH_*, are:

1. CH groups
(15)$$\left[1z\right]\overset{90{}^{\circ}xH}{\to }\overset{\emptyset {}^{\circ}yC}{\to }\overset{\delta_{1}}{\to }\overset{180{}^{\circ}xC}{\to }\overset{180{}^{\circ}xH}{\to }-\left[zx\right]\ldots \overset{90{}^{\circ}xC}{\to }-0.71ss^{\prime \prime }\left[x1\right]$$

2. CH_2_ groups
(16)$$\begin{array}{l} \left[1z1\right]+\left[11z\right]\overset{90{}^{\circ}xH}{\to }\overset{90{}^{\circ}xH^{\prime }}{\to }\overset{\emptyset {}^{\circ}yC}{\to }\overset{\delta_{1}}{\to }\overset{180{}^{\circ}xC}{\to }\overset{180{}^{\circ}xH}{\to }\overset{180{}^{\circ}xH^{\prime }}{\to }\ldots \overset{90{}^{\circ}xC}{\to }0.71sc^{\prime }s2^{\prime \prime }\left[x11\right]\\ -ss^{\prime }s^{\prime \prime 2}\left[x11\right] \end{array}$$

3. CH_3_ groups
(17)$$\begin{array}{l} \left[1z11\right]+\left[11z1\right]\\ +\left[111z\right]\overset{90{}^{\circ}xH}{\to }\overset{90{}^{\circ}xH^{\prime }}{\to }\overset{90{}^{\circ}xH^{\prime \prime }}{\to }\overset{\emptyset {}^{\circ}yC}{\to }\overset{\delta_{1}}{\to }\overset{180{}^{\circ}xC}{\to }\overset{180{}^{\circ}xH}{\to }\overset{180{}^{\circ}xH^{\prime }}{\to }\overset{180{}^{\circ}xH^{\prime \prime }}{\to }\ldots \overset{90{}^{\circ}xC}{\to }\\ -2.11sc^{\prime 2}s^{\prime \prime }c^{\prime \prime 2}\left[x111\right]+3sc^{\prime }s^{\prime }s^{\prime \prime 2}c^{\prime \prime }\left[x111\right]-1.06ss^{\prime 2}s^{\prime \prime 3}\left[x111\right] \end{array}$$

The final carbon signal intensities, represented by *I*, for the three groups CH, CH_2_, and CH_3_ for Ω = 90° and θ = 135°, as a function of the evolution delay δ and the actual coupling constant *J_CH_*, are thus:(18)*I_CH_* = *C_x_* [−0.71 sin(π*J_CH_*δ_1_) sin(π*J_CH_*δ_3_)]*I_CH2_* = *C_x_* [0.71 sin(π*J*_CH_δ_1_) cos(π*J*_CH_δ_2_) sin(2π*J*_CH_δ_3_) − 1.00 sin(π*J*_CH_δ_1_) sin(π*J*_CH_δ_2_) sin^2^(π*J*_CH_δ_3_)]*I_CH3_* = *C_x_* [−2.11 sin(π*J_CH_*δ_1_) cos^2^(π*J_CH_*δ_2_) sin(π*J_CH_*δ_3_) cos^2^(π*J_CH_*δ_3_) + 3 sin(π*J_CH_*δ_1_) cos(π*J_CH_*δ_2_) sin(π*J_CH_*δ_2_) cos(π*J_CH_*δ_3_) sin^2^(π*J_CH_*δ_3_) − 1.06 sin(π*J_CH_*δ_1_) sin^2^(π*J_CH_*δ_2_) sin^3^(π*J_CH_*δ_3_)]


DEPTQ^+^90_45_, Ω = 45°, θ = 90°

1. CH groups
(19)$$\begin{array}{l} \left[1z\right]\overset{45{}^{\circ}xH}{\to }\overset{\emptyset {}^{\circ}yC}{\to }\overset{\delta_{1}}{\to }\overset{180{}^{\circ}xC}{\to }\overset{180{}^{\circ}xH}{\to }-0.71s\left[zx\right]\overset{\left(90{}^{\circ}\right)xC}{\to }0.7s\left[yx\right]\overset{\left(180{}^{\circ}\right)xH}{\to }\overset{180{}^{\circ}xC}{\to }\overset{\delta_{2}}{\to }\\ -0.71s\left[yx\right]\overset{90{}^{\circ}yH}{\to }0.71s\left[yz\right]\overset{\left(180{}^{\circ}\right)xC}{\to }\overset{\left(180{}^{\circ}\right)xH}{\to }\overset{\delta }{\to }\overset{90{}^{\circ}xC}{\to }\\ -0.71ss^{\prime \prime }\left[x1\right] \end{array}$$

2. CH_2_ groups
(20)$$\left[1z1\right]+\left[11z\right]\overset{45{}^{\circ}xH}{\to }\overset{\emptyset {}^{\circ}yC}{\to }\overset{\delta_{1}}{\to }\overset{180{}^{\circ}xC}{\to }\overset{180{}^{\circ}xH}{\to }\overset{180{}^{\circ}xH^{\prime }}{\to }\ldots \overset{\delta }{\to }\overset{90{}^{\circ}xC}{\to }-0.71sc^{\prime }s2^{\prime \prime }[x11]$$

3. CH_3_ groups
(21)$$\begin{array}{l} \left[1z11\right]+\left[11z1\right]+\left[111z\right]\overset{45{}^{\circ}xH}{\to }\overset{\emptyset {}^{\circ}yC}{\to }\overset{\delta_{1}}{\to }\overset{180{}^{\circ}xC}{\to }\overset{180{}^{\circ}xH}{\to }\overset{180{}^{\circ}xH^{\prime }}{\to }\ldots \overset{\delta }{\to }\overset{90{}^{\circ}xC}{\to }\\ -2.11sc^{\prime 2}s^{\prime \prime }c^{\prime \prime 2}\left[x111\right] \end{array}$$

The final carbon signal intensities, represented by *I*, for the three groups CH, CH_2_, and CH_3_ for Ω = 45° and θ = 90°, as a function of the three evolution delays δ_1_, δ_2_ and δ_3_ and the actual coupling *J_CH_*, are therefore:(22)*I_CH_* = −*C_x_* [0.71 sin (π*J_CH_*δ_1_) sin (π*J_CH_*δ_3_)]*I_CH2_* = −*C_x_* [0.71 sin (π*J*_CH_δ_1_) cos (π*J*_CH_δ_2_) sin (2π*J*_CH_δ_3_)]*I_CH3_* = −*C_x_* [2.11 sin (π*J*_CH_δ_1_) cos^2^(π*J*_CH_δ_2_) sin (π*J*_CH_δ_3_) cos^2^(π*J*_CH_δ_3_)]


For the DEPTQ and DEPTQ^+^ experiments, the approximate carbon signal intensities, represented by *I* (for ^1^*J_CH_ ≈* ^1^*J_CH_^0^*), for the three groups CH, CH_2_, and CH_3_ for Ω = 90° and θ = 135°, as a function of the coupling constant ^1^*J_CH_* and δ, δ_1_, δ_2_, and δ_3_, are summarized in Table 1:

According to the above relations for Ω = 90° and θ = 135° and compared to the DEPTQ experiment, the signs of the signal intensities for CH and CH_3_ groups are *inverted* using the DEPTQ^+^ experiment, while the sign of the signal intensities for CH_2_ groups is *identical*. Consequently, the usual angle θ for achieving the DEPT editing, CH/CH_3_ positive, CH_2_ negative, is obtained for θ = 45° and not for θ = 135°, as would be valid for DEPTQ (Table 2).

To separate the signals of CH, C_q_ and CH_2_, CH_3_ groups in different sub-spectra and to obtain different signs for the signals of CH_2_ and CH_3_ groups the first adjustable proton pulse Ω and the editing pulse θ must therefore be set for the DEPTQ^+^ experiment to 45° and 90° (DEPTQ^+^90_45_) and to 90° and 45°, respectively (DEPTQ^+^45_90_). Subtracting the DEPTQ^+^90_45_ spectrum, showing exclusively the Cq and CH signals with opposite phases, from the DEPTQ^+^45_90_ spectrum results in a CH_2_/CH_3_ edited spectrum with the signals of the two carbon types with opposite phases.

Interestingly, equations 15–22 show that the intensities of the signals of the three carbon types depend differently and for CH *solely* on *two* (δ_1_ and δ_3_) of the individual delays δ_1_, δ_2_, and δ_3_, which is the basis for efficiently leveling the signal intensities for all multiplicities with the DEPTQ^+^ pulse sequence, even for large ^1^*J*_CH_ coupling constant ranges. Indeed, and with the specific ranges of ^1^*J*_CH_-coupling constants for the three carbon types, i.e., CH: [120 < ^1^*J*_CH_ < 250 Hz], CH_2_: [115 < ^1^*J*_CH_ < 175 Hz], CH_3_: [100 < ^1^*J*_CH_ < 135 Hz], the three delays δ_1_, δ_2_, and δ_3_ can be individually and optimally adjusted. For instance, δ_2_ which does not affect CH-intensities can be adjusted for small ^1^J_CH_ coupling constants, typically 120 Hz, to maximize the signals of CH_2_ and CH_3_ groups. On the other hand, δ_1_ can be adjusted for large ^1^*J*_CH_ coupling constants, typically 220–230 Hz, to maximize the signals of acetylenic CH groups. Finally, δ_3_ can be adjusted for an average ^1^*J*_CH_ coupling constant, typically 185 Hz, to account for all carbon types present. This enables real leveling of the signal intensities with the DEPTQ^+^ experiment, as demonstrated theoretically (Appendix A), and, in the following section, practically.

### 2.2. Experimental Results

#### 2.2.1. Cholesteryl Acetate

We first used a sample of 100 mmol cholesteryl acetate (Figure 3, 30 mg dissolved in 0.7 mL CDCl_3_) for testing the performance of the DEPTQ^+^ experiment.

In Figure 4 and Figure 5, the DEPTQ135_90_, DEPTQ90_45_, the difference between DEPTQ135_90_ and DEPTQ90_45_ (Figure 4), the DEPTQ^+^45_90_, DEPTQ^+^90_45_, and the difference between DEPTQ^+^45_90_ and DEPTQ^+^90_45_ (Figure 5) of cholesteryl acetate are shown. Cholesteryl acetate (Figure 3) is a molecule exhibiting a narrow range of coupling constants ^1^*J*_CH_ (122 Hz for ^1^*J*_C26H26_ to 155 Hz for ^1^*J*_C6H6_). Accordingly, both the DEPTQ90_45_ providing a Cq/CH edited spectrum and the DEPTQ90_45_ providing a CH_2_/CH_3_ edited spectrum (difference between DEPTQ135_90_ and DEPTQ90_45_) are of high quality, with few and weak CH_2_/CH_3_ artefacts in the DEPTQ90_45_ and some residual CH peaks in the difference spectrum, respectively (Figure 4).

The same, nearly identical results have been obtained using the proposed DEPTQ^+^ pulse sequence. Thus, the DEPTQ^+^ behaves similarly to the DEPTQ experiment if the narrow, usual range of coupling constants ^1^*J*_CH_ is present: the mutual signal intensity leveling and the *J* cross-talk reduction only provide very minor improvements. 

However, the superiority of the DEPTQ^+^ experiment becomes clear when there is a larger range of coupling constants ^1^*J*_CH_. First, with the sole aim of demonstrating the potential and performance of the DEPTQ^+^ experiment, we used the same molecule of cholesteryl acetate but adjusted the three delays in the experiments as if all ^1^*J*_CH_ coupling constants (alkanes, aromatics, alkynes) were present. The spectra are shown in Appendix A.

In this case, it is clear that the DEPTQ^+^ behaves much better than the DEPTQ experiment; even the standard DEPTQ135 spectrum is confusing, since the resonances of the methyl groups between 10 and 30 ppm are barely visible (Appendix A), as predicted theoretically (Appendix A). This is due to the very large mismatch between the actual ^1^*J*_CH_ coupling constants in methyl groups (~125 Hz) and the average ^1^*J*_CH_^0^ coupling constant set in the DEPTQ experiment, ^1^*J*_CH_^0^ = 185 Hz, intentionally set to cover the whole range of ^1^*J*_CH_ coupling constants (100–250 Hz). The DEPTQ90_45_ is also confusing, with many residual CH_2_/CH_3_ signals present, which makes the identification of CH groups difficult, as they are as intense (strong *J* cross talk) and have the same sign as the CH signals (Appendix A). Finally, the “CH_2_/CH_3_” edited spectrum (difference between DEPTQ135_90_ and DEPTQ90_45_) is extremely confusing, as only the resonances of CH_2_ groups are present, while the resonances of CH_3_ groups are either barely visible or even completely absent (Appendix A). 

In contrast, the spectra obtained using the DEPTQ^+^ pulse sequence are much better and sufficient for reliable identification of CH_n_ groups: In the DEPTQ^+^45_90_ spectrum, the resonances of the methyl groups between 10 and 30 ppm are clearly visible and intense (Appendix A), as predicted theoretically (Appendix A). This is because the three delays δ_1_, δ_2_, and δ_3_ in the DEPTQ^+^ experiment are matched to three different ^1^*J*_CH_^0^ coupling constants (230, 125, 165 Hz, respectively), which results in a mutual leveling of the DEPT signal intensities. In the DEPTQ^+^90_45_, there are only very few and weak CH_2_/CH_3_ artefacts (weak *J* cross talk), which enables easy identification of CH groups (Appendix A). Finally, the “CH_2_/CH_3_” edited spectrum also allows a reliable identification of the clearly visible resonances of CH_2_ groups and of CH_3_ (Appendix A). 

#### 2.2.2. 4-Methyl-*N,N*-di(prop-2-yn-1-yl)aniline

In order to test the effective performance, we finally tried the DEPTQ^+^ experiment on the demanding sample of ~30 mg of 4-methyl-*N,N*-di(prop-2-yn-1-yl)aniline dissolved in 0.7 mL CDCl_3_ (Figure 1). 4-methyl-*N,N*-di(prop-2-yn-1-yl)aniline contains alkane, aromatic and alkyne groups with the full range of ^1^*J*_CH_ coupling constants (125 Hz for ^1^*J*_C8H8_ to 248 Hz for ^1^*J*_C1H1_). Note also the very particular case of C2 at 79.4 ppm, which, with its large long-range coupling (^2^*J*_C2H1_ ~ 50 Hz), behaves like a pseudo- or only partially quaternary carbon.

In Figure 6 and Figure 7, the DEPTQ135_90_, DEPTQ90_45_, the difference between DEPTQ135_90_ and DEPTQ90_45_ (Figure 6), and the DEPTQ^+^45_90_, DEPTQ^+^90_45_, and the difference between DEPTQ^+^45_90_ and DEPTQ^+^90_45_ (Figure 7) of 4-methyl-*N,N*-di(prop-2-yn-1-yl)aniline are shown. 

With 4-methyl-*N,N*-di(prop-2-yn-1-yl)aniline, it is clear that the DEPTQ^+^ performs much better than the DEPTQ experiment: the standard DEPTQ135 spectrum is useable, but the resonance of the methyl group C8 at 20.6 ppm is very weak (Figure 6). As already mentioned, this is due to the very large mismatch between the actual ^1^*J*_CH_ coupling constants in methyl groups (~125 Hz) and the large average ^1^*J*_CH_^0^ coupling constant set in the DEPTQ experiment, ^1^*J*_CH_^0^ = 185 Hz (Appendix A). Artefacts (*J* cross talk) belonging to the CH_2_ C3 at 40.9 ppm and to the CH_3_ C8 at 20.6 ppm are visible in the DEPTQ90_45_, even if their intensities are rather small (Figure 6). Such artefacts can be problematic with unknown molecular structures or if the attribution is performed in an automatic or semi-automatic way using dedicated attribution software. Particularly confusing is the “CH_2_/CH_3_” edited spectrum obtained with DEPTQ (difference between DEPTQ135_90_ and DEPTQ90_45_), with a strong signal of the CH_2_ groups C3 at 40.9 ppm and a weak residual signal of the CH group C1, while the resonance of the CH_3_ group C8 at 20.6 ppm is almost invisible (Figure 6). In addition, the residual signal of C1 at 72.8 ppm could be misleading or misinterpreted by attribution/analysis software.

In contrast, the spectra obtained using the DEPTQ^+^ pulse sequence are less confusing and sufficient for reliable carbon type identification: in the DEPTQ^+^45_90_, the resonance of the methyl group C8 around 20 ppm is clearly visible and intense (Figure 7). This must be attributed to the individual tuning of the *three* delays to different ^1^*J*_CH_^0^ coupling constants (125, 175, 230 Hz), which results in a mutual leveling of the DEPT signal intensities. The DEPTQ^+^90_45_ shows—besides the negative quaternary carbon signals—the three strong CH signals with only one weak CH_2_ artifact (weak *J* cross talk) present (Figure 7). Finally, the CH_2_/CH_3_ edited spectrum is almost perfect, with the strong resonances of the CH_2_- and CH_3_ groups C3 and C8 in anti-phase and only very weak residual signals left (Figure 7). 

### 2.3. Practical Aspects

Our theoretical and practical investigations show that the proposed new DEPTQ^+^45_90_ and DEPTQ^+^90_45_ experiments work well. However, the experimental results suggest that the quality of spectral editing also depends on the relaxation delay. While for medium sized molecules such as cholesteryl acetate, the length of the relaxation delay affects only the intensity of the slow relaxing carbons but not the editing quality (Appendix A), it becomes critical also for the editing procedure for small molecules and/or groups exhibiting long T_1_ relaxation times.

This manifests itself with 4-methyl-*N,N*-di(prop-2-yn-1-yl)aniline, for which the intensities of the CH signals in the DEPTQ^+^90_45_ spectra can be drastically different to that obtained in the DEPTQ^+^45_90_ if overly short relaxation delays are used. The numerous and intense “CH-artefacts” showing up in the edited CH_2_/CH_3_ spectrum may be potentially attributed to CH_3_ groups. The influence of relaxation and the need for choosing the relaxation delay(s) long enough with DETPQ^+^ to achieve a sufficient editing quality are corroborated by simulated spectra, which are in good agreement with the corresponding experimental spectra shown in Appendix A. 

Finally, we wish to concentrate on the overall sensitivities of DEPTQ^+^, and to compare it with DEPTQ and the usually performed “tandem alternatives” such as ^13^C one-pulse/DEPT. In previous reports [9,10,11], Bigler et al. have compared spectra acquired DEPTQ and the ^13^C one-pulse spectra with an equal number of scans within the same total measuring time, and found that the DEPTQ spectrum decreases signal intensities of quaternary carbons by 15% and 10% when compared to the ^13^C one-pulse. The corresponding experiments (DEPTQ vs. ^13^C one-pulse + DEPT) based on equal total measuring times showed average S/N ratios of about 70% and 85% for the CH_n_ signals in the DEPT spectrum, compared to the corresponding values of the DEPTQ experiment [10]. While the signal intensities of quaternary carbons are identical in DEPTQ and DEPTQ^+^ spectra, the comparison of the average S/N ratio for the CH_n_ signals between DEPTQ and DEPTQ^+^ is not so straightforward. While for standard DEPTQ experiments, as shown in this manuscript, the signals at the extremes of the ^1^*J*_CH_ coupling range suffer in intensity or cannot be detected, DEPTQ^+^ experiments (similar to ACCORD-DEPT experiments [31]) provide spectra with notably improved S/N ratios for these specific signals. However, owing to the mutual signal intensity leveling, DEPTQ^+^ always provides either better or worse signal intensity compared to standard optimization. The S/N ratio for some signals will correspondingly decrease compared to a standard optimized DEPTQ experiment. 

## 3. Materials and Methods

### 3.1. Synthesis and Characterization of 4-Methyl-N,N-di(prop-2-yn-1-yl)aniline

The chemicals were purchased from Aldrich, Alfa Aesar, Acros Organics, and TCI Chemicals and used without further purification, In a flame-dried round bottom flask equipped with a magnetic stirrer, *p*-toluidine (0.5 g, 4.6 mmol) and anhydrous K_2_CO_3_ (0.77 g, 5.52 mmol) were mixed in dry dimethylformamide (10 mL). Propargyl bromide (0.65 g, 5.52 mmol) was then added dropwise during a period of 15 min and the resulting mixture was stirred under nitrogen atmosphere for 5 h. After completion of the reaction (monitored by TLC), the mixture was poured into water (10 mL) and extracted with ethyl acetate (2 × 15 mL). The combined organic layers were washed with water (4 × 10 mL) and brine (10 mL), dried over sodium sulfate, filtered, and concentrated under reduced pressure [32,33]. The reaction crude was purified by column chromatography (Silica, 230–400 mesh) using mixtures of hexane/ethyl acetate (30:1). The desired product was obtained as a highly dense yellow oil (110 mg, 13%) along with the product of mono substitution 4-methyl-*N*-(prop-2-yn-1-yl)aniline (500 mg, 75%). 

*4-methyl-N,N-di(prop-2-yn-1-yl)aniline:*^1^H NMR (300 MHz, CDCl_3_) δ 7.12 (d, *J* = 8.6 Hz, 2H), 6.92 (d, *J* = 8.6 Hz, 2H), 4.11 (d, *J* = 2.4 Hz, 4H), 2.30 (s, 3H), 2.26 (t, *J* = 2.4 Hz, 2H). ^13^C NMR (75 MHz, CDCl_3_) δ 145.8 (C4), 129.8 (C5), 129.7 (C7), 116.7 (C6), 79.4 (C2), 72.8 (C1), 40.9 (C3), 20.6 (C8). The spectra are referenced with respect to the residual CHCl_3_ resonance (7.264 ppm) and to the ^13^C resonance of CDCl_3_ (77.16 ppm). 

### 3.2. NMR Measurements

All DEPTQ and DEPTQ^+^ experiments were recorded on a Bruker AvanceII 500 MHz NMR spectrometer equipped with a 5 mm BBI Inverse probehead. The samples used were: (i) the standard test sample provided with Bruker NMR spectrometers, 30 mg of cholesteryl acetate dissolved in 0.7 mL CDCl_3_, and (ii) ~30 mg of 4-methyl-*N,N*-di(prop-2-yn-1-yl)aniline dissolved in 0.7 mL CDCl_3_. ^1^H and ^13^C 90° pulse lengths were 8.4 μs and 13.2 μs, respectively. The composite chirp pulse for refocusing has a duration of 2 ms, is defined by 4000 points and by 20% smoothing and 60 kHz sweep width (Crp60comp.4 in the Bruker wave form library). The experimental details of the measurements are as follows. Cholesteryl acetate: if not otherwise mentioned, all spectra were acquired with 128 k real data points with a relaxation delay of 4 s, a NOE building period of 3 s, 64 scans for a total experimental time of approximately 7 min. ^13^C spectral widths were 200 ppm, leading to an acquisition time of 2.60 s. Data were processed with 128 k data points using exponential multiplication with a line broadening of 1 Hz. 4-methyl-*N,N*-di(prop-2-yn-1-yl)aniline: the spectra were acquired with 128 k data points with a relaxation delay of 10 s, a NOE building period of 5 s, 16 scans for a total experimental time of approximately 3 min. ^13^C spectral widths were 200 ppm, leading to an acquisition time of 2.60 s. Data were processed with 128 k data points using exponential multiplication with a line broadening of 1 Hz.

### 3.3. Numerical Simulations

The numerical simulations shown in Appendix A have been performed with Microsoft Excel^®^, Office 365 for Windows. Relaxation effects during both pulse sequences were not considered.

### 3.4. NMR Simulations

The simulations shown in Appendix A have been performed with the BRUKER NMRSIM program for MAC (version 5.5.3. 2012). The simulated DEPTQ and DEPTQ^+^ spectra are obtained for 4-methyl-*N,N*-di(prop-2-yn-1-yl)aniline. The parameters used for the simulations were: ^1^*J*_C1H1_ = 260 Hz, ^1^*J*_C3H3_ = 145 Hz, ^1^*J*_C5H5_ = ^1^*J*_C6H6_ = 165 Hz, ^1^*J*_C8H8_ = 125 Hz, ^2^*J*_C2H1_ = 50 Hz. The relaxation times have been set to T_1_ = 1 s for all CH_n_ groups, and T_1_ = 5 s for the quaternary carbons, T_2_ = 0.5 s, and full relaxation during both the pulse sequence and final data acquisition were considered. The chemical shifts set in the simulations were the experimental chemical shifts.

## 4. Conclusions

Compared to the DEPTQ experiment, the proposed new DEPTQ^+^ experiment is shorter and the different evolution delays are designed as spin echoes, which can be matched to different ^1^*J*_CH_ values. We have shown, both theoretically and experimentally, that for molecules with a standard range of ^1^*J*_CH_ coupling constants (i.e., 110,180 Hz), the differences between DEPTQ and DEPTQ^+^ are minimal. The differences become appreciable for molecules possessing a large range of ^1^*J*_CH_ coupling constants. The DEPTQ^+^ experiment makes it possible to obtain DEPTQ spectra with (i) a mutual leveling of the DEPT signal intensities, (ii) a reduction in *J* cross-talk in the Cq/CH spectrum, and (iii) more reliable CH_2_/CH_3_ edited spectra. 

The influence of relaxation and the need to choose a relaxation delay that is sufficiently long, with DETPQ^+^, to obtain reliable data and clean edited spectra represents the only limitation of this study, but this is valid for molecules with long T_1_ relaxation times. This is, however, also the case with the original DEPTQ experiment.

In conclusion, the new DEPTQ^+^ experiment is expected to be attractive for fast ^13^C analysis of small-to medium sized molecules, especially those with a large range of ^1^*J*_CH_ coupling constant, and especially in high-throughput laboratories. With concentrated samples and/or by exploiting the high sensitivity of cryogenically cooled ^13^C NMR probeheads, the efficacy of such investigations may be improved, as it is possible to unequivocally identify all carbon multiplicities with only one scan for each of the two independent DEPTQ^+^ experiments.

## Figures and Tables

**Figure 1 molecules-26-03490-f001:**
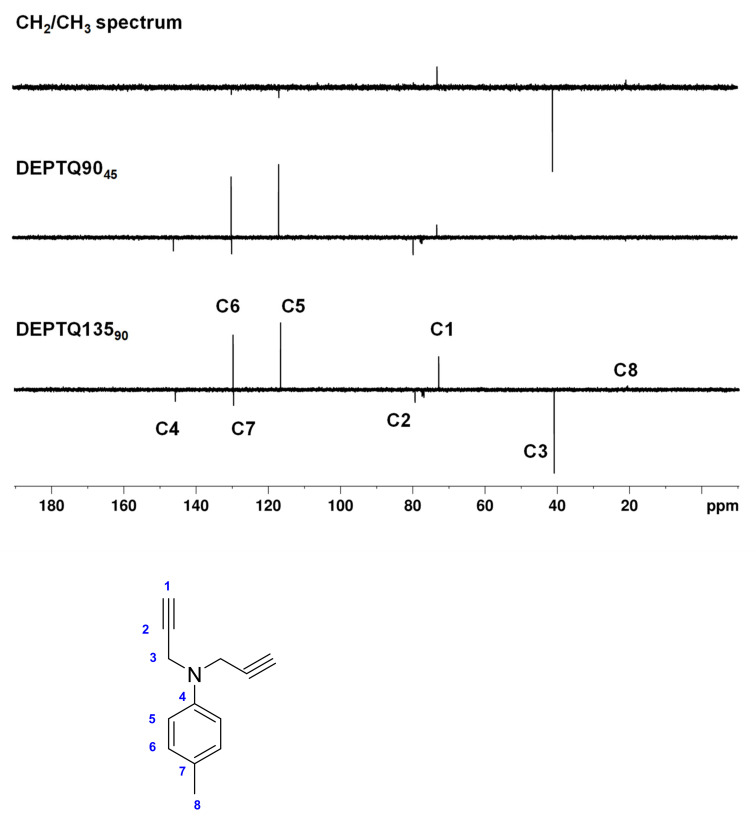
4-methyl-*N,N*-di(prop-2-yn-1-yl)aniline and carbon numbering and DEPTQ135_90_, DEPTQ90_45_, and the difference between DEPTQ135_90_ and DEPTQ90_45_ spectra of ~30 mg of 4-methyl-*N,N*-di(prop-2-yn-1-yl)aniline dissolved in 0.7 mL CDCl_3_. The delay δ was set to 2.70 ms, adjusted for a coupling constant ^1^*J*_CH_ of 185 Hz. The relaxation delay was 2 s, the NOE building period 1 s. All other parameters are identical to those described in the Materials and Methods section.

**Figure 2 molecules-26-03490-f002:**
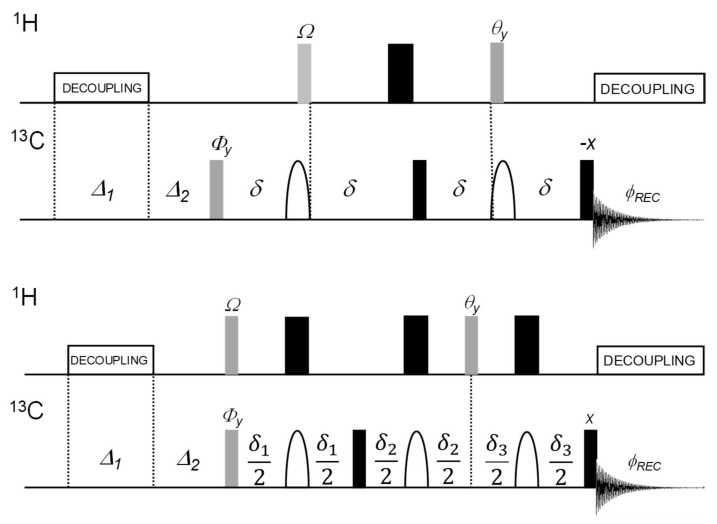
DEPTQ [11] (**top**) and DEPTQ^+^ pulse sequences (**bottom**) for editing CH_n_ multiplicities. The first adjustable proton pulse Ω and the editing pulse θ are set to 45° and 90^o^, respectively, for a DEPTQ90 (DEPTQ90_45_) and to 90° and 135°, respectively, for a DEPTQ135 (DEPTQ135_90_), and the delay δ is adjusted to 1/(2 * ^1^*J*_CH_). The DEPTQ90_45_ spectrum, showing exclusively the Cq and CH signals with opposite phases, and the DEPTQ135_90_ spectrum, are subtracted from each other results in a CH_2_/CH_3_ edited spectrum with the signals of the two carbon types with opposite phases [11]. The ^13^C 180° (refocusing) pulses are composite smoothed chirp pulses (2 ms total duration, 60 kHz sweep width). The initial ^13^C 90° pulse may be replaced by a variable φ pulse, adjustable for maximum sensitivity (Ernst angle). An additional 90° ^13^C pulse prior to data acquisition re-establishes the residual ^13^C-*z* magnetization left with the initial ^13^C pulse. In the DEPTQ^+^, compared to the original DEPTQ version [10,11], the first evolution delay δ is omitted, and the remaining three evolution periods are defined as spin echoes, thus allowing different values for δ_1_, δ_2_, and δ_3_ to be used. The first adjustable proton pulse Ω and the editing pulse θ are set to 45° and 90°, respectively, for a DEPTQ^+^90 (DEPTQ^+^90_45_) and to 90° and 45^o^, respectively, for a DEPTQ^+^45 (DEPTQ^+^45_90_).

**Figure 3 molecules-26-03490-f003:**
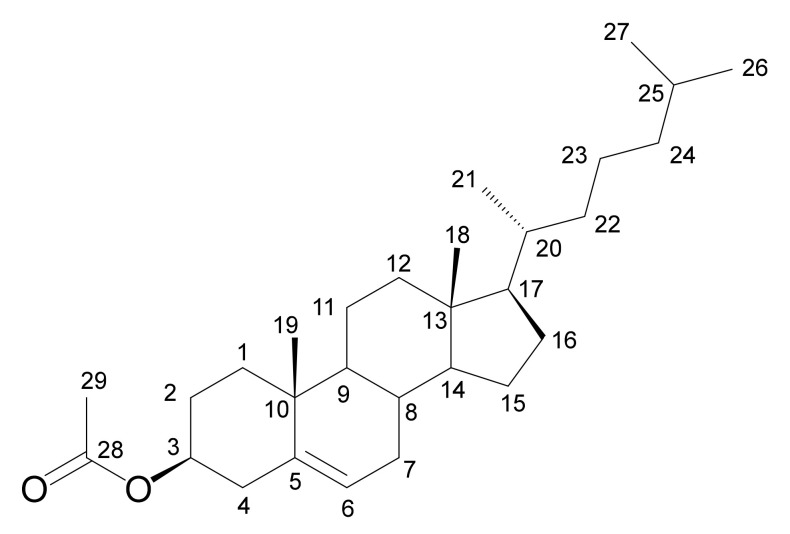
Cholesteryl acetate and carbon numbering.

**Figure 4 molecules-26-03490-f004:**
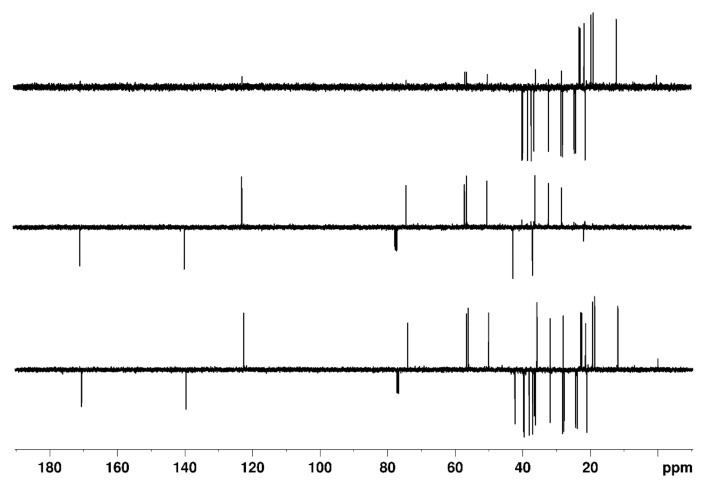
**Bottom**: DEPTQ135_90_; **middle**: DEPTQ90_45_; **top**: Difference between DEPTQ135_90_ and DEPTQ90_45_ spectra of 30 mg of cholesteryl acetate dissolved in 0.7 mL CDCl_3_. The delay δ was set to 3.45 ms, adjusted for a coupling constant ^1^*J*_CH_ of 145 Hz.

**Figure 5 molecules-26-03490-f005:**
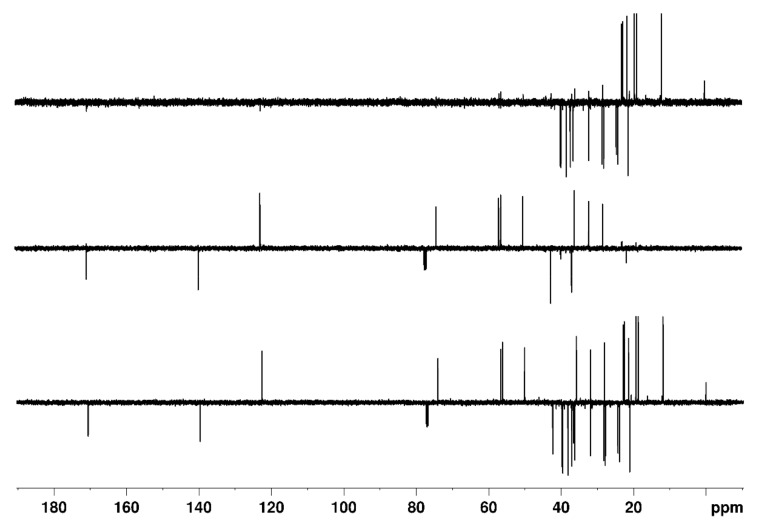
**Bottom**: DEPTQ^+^45_90_; **middle**: DEPTQ^+^90_45_; **top**: Difference between DEPTQ^+^45_90_ and DEPTQ^+^90_45_ spectra of 30 mg of cholesteryl acetate dissolved in 0.7 mL CDCl_3_. The delays δ_1_, δ_2_, and δ_3_ were set to 3.13 ms, adjusted for a coupling constant ^1^*J*_CH_ of 160 Hz, to 4.35 ms, adjusted for a coupling constant ^1^*J*_CH_ of 115 Hz, and to 3.45 ms, adjusted for a coupling constant ^1^*J*_CH_ of 145 Hz, respectively.

**Figure 6 molecules-26-03490-f006:**
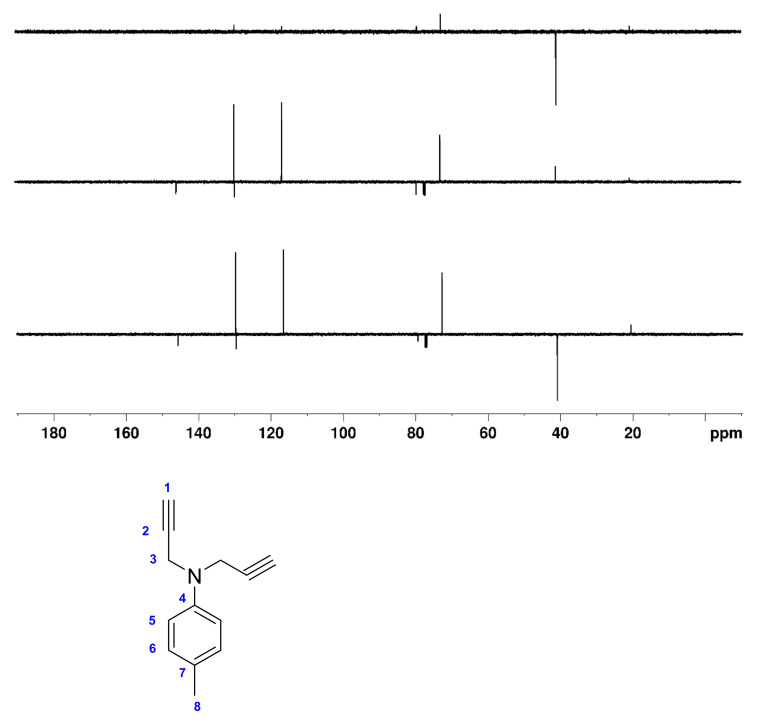
**Bottom**: DEPTQ135_90_, **middle**: DEPTQ90_45_; **top**: Difference between DEPTQ135_90_ and DEPTQ90_45_ spectra of ~30 mg of 4-methyl-*N,N*-di(prop-2-yn-1-yl)aniline dissolved in 0.7 mL CDCl_3_. The delay δ was set to 2.70 ms, adjusted for a coupling constant ^1^*J*_CH_ of 185 Hz. The relaxation delay was 4 s, the NOE building period 3 s. All other parameters are identical to those described in the Materials and Methods section. The structure of 4-methyl-*N,N*-di(prop-2-yn-1-yl)aniline and carbon numbering is shown below the spectra.

**Figure 7 molecules-26-03490-f007:**
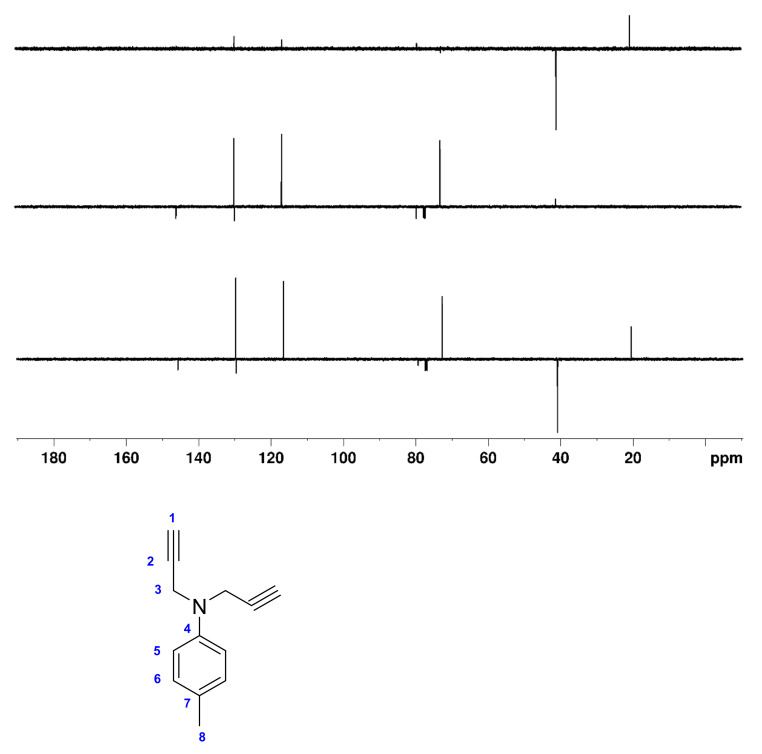
**Bottom**: DEPTQ^+^45_90_; **middle**: DEPTQ^+^90_45_; **top**: Difference between DEPTQ^+^45_90_ and DEPTQ^+^90_45_ spectra of ~30 mg of 4-methyl-*N,N*-di(prop-2-yn-1-yl)aniline dissolved in 0.7 mL CDCl_3_. The delays δ_1_, δ_2_, and δ_3_ were set to 2.18 ms, adjusted for a coupling constant ^1^*J*_CH_ of 230 Hz, to 4.00 ms, adjusted for a coupling constant ^1^*J*_CH_ of 125 Hz, and to 3.18 ms, adjusted for a coupling constant ^1^*J*_CH_ of 165 Hz, respectively. The relaxation delay was 4 s, the NOE building period 3s. All other parameters are identical to those described in the Materials and Methods section. The structure of 4-methyl-*N,N*-di(prop-2-yn-1-yl)aniline and carbon numbering is shown below the spectra.

**Table 1 molecules-26-03490-t001:** Final carbon signal intensities, represented by *I*, for the three groups CH, CH_2_, and CH_3_ for Ω = 90°, θ = 135°, as a function of the coupling constant *J_CH_* and the delays δ, δ_1_, δ_2_, and δ_3_.

	DEPTQ	DEPTQ^+^
CH groups	$0.71C_{x}$sin^2^(π*J_CH_*δ)	$-0.71C_{x}$sin(π*J_CH_*δ_1_) sin(π*J_CH_*δ_3_)
CH_2_ groups	$-C_{x}$sin^4^(π*J_CH_*δ)	$-C_{x}$sin(π*J_CH_*δ_1_) sin(π*J*_CH_δ_2_) sin^2^(π*J*_CH_δ_3_)
CH_3_ groups	$1.06C_{x}$sin^6^(π*J_CH_*δ)	$-1.06C_{x}$sin(π*J*_CH_δ_1_) sin^2^(π*J*_CH_δ_2_) sin^3^(π*J*_CH_δ_3_)

**Table 2 molecules-26-03490-t002:** DEPTQ and DEPTQ^+^: maximal amplitude of CH, CH_2_, and CH_3_ groups for the three usual angles θ and for Ω = 90°.

Experiment	Angle θ	Amplitude of CH	Amplitude of CH_2_	Amplitude of CH_3_
DEPTQ^+^	45°	−0.71	1	−1.06
DEPTQ		0.71	1	1.06
DEPTQ^+^	90°	−1	0	0
DEPTQ		1	0	0
DEPTQ^+^	135°	−0.71	−1	−1.06
DEPTQ		0.71	−1	1.06

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
