# Peer review of "The DEPTQ+ Experiment: Leveling the DEPT Signal Intensities and Clean Spectral Editing for Determining CHn Multiplicities"

_molecules, 2021, doi:10.3390/molecules26123490_

Round 1

Reviewer 1 Report

The authors proposed a new pulse sequence DEPTQ+ which is suitable for DEPT editing spectra. There is clear improvement in comparison to DEPTQ experiments. Theoretical parts are important and suitable for NMR specialists, but it may be difficult to understand for ordinary NMR users using for routine NMR measurements.  Some of parts could be moved to Supporting information.

Relatively long relaxation time and NOE time are used in DEPTQ+ experiments.

It is obviously useful for high concentration sample and NMR with high sensitive cryogenic probe head. The reviewer recommended that it will be appreciated that addition of discussion on S/N ratio using recommended parameters such as adjusted for 230, 125 and 165 Hz, and in comparison with optimized parameter for cholesteryl acetate, and together with relaxation time and repetition time or total experimental time, especially for Cq intensities along with 13C single pulse experiments. It is important to appeal the proposed DEPTQ+ for ordinary NMR users who do not have NMR with cryogenic probe head. Other comments are summarized to below for consideration.

P2L15 We also observed that that  => We also observed that

Figure 1. C5 and C6 assignments should be exchanged on DEPTQ13590spectrum.

4-methyl-N,N-di(prop-2-yn-1-yl)aniline.   N,N is initalic

Figure 2 fRECis italic or not. Please make it the same for both pulse sequence.

P4L1 and Figure 2, and so on.   0.5/1JCH    => 1/21JCH

P5 eq.(1) Please change line break positions.

-cosf[z] +sinf[x]

cosf[y] +sinf[x]

cosf[z] +sinf[x]     f is phi in Greek

And other equations, please check them.

P6L6 furanyl groups, for which the 1JCH~ 210-250Hz

1JCH  values for furanare ca 202 and 175Hz.

P6L7(ii) for methyl groups (1JCH~ 100-130 Hz)

1JCH  values  for typical CH3-R (R = alkanes, alkenes, and alkynes) are ca 123 – 132 Hz. 1JCH  value ca 100Hz will be match with CH3-Li is 98Hz, which will be special case. Other special cases are well known for CH3-X (X=F, CL, Br, and I)

and those 1JCH  values are ca 150Hz.

Please make some modifications. For example, furanyl groups could be replaced by hetero aromatic rings.

P10L13 CH2:[115 < 1JCH< 150Hz] 

Range of 1JCH for CH2 groups is narrow, because there are many large values, e.g. formaldehyde 1JCH=172Hz, ethylene oxide1JCH=176Hz, and ethylene 1JCH=156Hz.

Figure 3. Beta stereochemistry for acetate on C-3 should be added.

Numbering should be normal cholesterol numbering. 6-29 should be replaced by 4-27, and also 4 and 5 by 28 and 29.

P11L4 1JCH(112Hz for 1JC28H28  J value for terminal methyl groups should be larger than 120 Hz. Please check the values. Numbering should be also revised based on the revision of Figure 3.

P12L11 methyl groups (~110Hz) => methyl groups (~125Hz)

P13L11 115Hz for 1JC8H8    Is it a real 1JCHvalue?  1JCHvalue for methyl group of toluene is 126Hz. Please check it.

Author Response

Rev 1

Comments and Suggestions for Authors

The authors proposed a new pulse sequence DEPTQ+ which is suitable for DEPT editing spectra. There is clear improvement in comparison to DEPTQ experiments. Theoretical parts are important and suitable for NMR specialists, but it may be difficult to understand for ordinary NMR users using for routine NMR measurements.  Some of parts could be moved to Supporting information.

  • We understand the point of view of this referee. However, as underlined by this referee, theoretical parts are important and suitable for NMR specialists, and we therefore think that the whole development / explanations should be part of the main text. Other readers can skip those parts without any detrimental and concentrate on the experimental results.

Relatively long relaxation time and NOE time are used in DEPTQ+ experiments.

It is obviously useful for high concentration sample and NMR with high sensitive cryogenic probe head. The reviewer recommended that it will be appreciated that addition of discussion on S/N ratio using recommended parameters such as adjusted for 230, 125 and 165 Hz, and in comparison with optimized parameter for cholesteryl acetate, and together with relaxation time and repetition time or total experimental time, especially for Cq intensities along with 13C single pulse experiments. It is important to appeal the proposed DEPTQ+ for ordinary NMR users who do not have NMR with cryogenic probe head.

  • We thank this reviewer for the advice. The comparison DEPTQ/13C one pulse experiment has already been extensively done in previous publications from one of the corresponding authors (P Bigler, Magn. Reson. 1998, 135, 529-534, Magn. Reson. Chem. 2007, 45, 469-472, Spectrosc. Lett. 2008, 41, 162-165.). We have added a few sentences at the end of the practical considerations paragraph summarizing these findings.

Other comments are summarized to below for consideration.

P2L15 We also observed that that  => We also observed that

  • Second that deleted

Figure 1. C5 and C6 assignments should be exchanged on DEPTQ13590spectrum.

  • Thank you for having noticed this inattention. The assignments have been exchanged.

4-methyl-N,N-di(prop-2-yn-1-yl)aniline.   N,N is initalic

  • Yes, N,N was changed to N,N everywhere

Figure 2 fRECis italic or not. Please make it the same for both pulse sequence.

  • Thank you for having noticed this inconsistency. We have used this opportunity to further modify Fig 2 and put all letters (f, d, etc in italic)

P4L1 and Figure 2, and so on.   0.5/1JCH    => 1/21JCH

  • One of the corresponding author (J. Furrer) is used with this notation (5/1JCH ) for almost 20 years. We however agree that for this particular publication in Molecules, 1/(2 * 1JCH) is more suitable and comprehensible. We therefore changed 0.5/1JCH    to 1/(2 * 1JCH) everywhere in the text.

P5 eq.(1) Please change line break positions.

-cosf[z] +sinf[x]

cosf[y] +sinf[x]

cosf[z] +sinf[x]     f is phi in Greek

And other equations, please check them.

  • We are sorry, but we are unable to improve the aspect of the equations: Microsoft equation is not flexible and add line breaks randomly. May be the editing team can further help?

P6L6 furanyl groups, for which the 1JCH~ 210-250Hz

1JCH  values for furanare ca 202 and 175Hz.

  • We have changed this. However, depending on the sources used (we guess this referee refers to the fantastic database of H. Reich, (https://organicchemistrydata.org/hansreich/resources/nmr/?index=nmr_index%2F13C_coupling#c-coupling24), 1JCH~ 175-210Hz for furanyl (J.Org.Chem.2015, 80, 10838−10848)

P6L7(ii) for methyl groups (1JCH~ 100-130 Hz)

1JCH  values  for typical CH3-R (R = alkanes, alkenes, and alkynes) are ca 123 – 132 Hz. 1JCH  value ca 100Hz will be match with CH3-Li is 98Hz, which will be special case. Other special cases are well known for CH3-X (X=F, CL, Br, and I)

and those 1JCH  values are ca 150Hz.

  • We have modified to : for typical methyl groups (1JCH~ 110-135 Hz)

Please make some modifications. For example, furanyl groups could be replaced by hetero aromatic rings.

  • Indeed, this was modified

P10L13 CH2:[115 < 1JCH< 150Hz] 

Range of 1JCH for CH2 groups is narrow, because there are many large values, e.g. formaldehyde 1JCH=172Hz, ethylene oxide1JCH=176Hz, and ethylene 1JCH=156Hz.

  • We modified to: typical ethylene CH2 groups :[115 < 1JCH< 175Hz]

Figure 3. Beta stereochemistry for acetate on C-3 should be added.

  • Added

Numbering should be normal cholesterol numbering. 6-29 should be replaced by 4-27, and also 4 and 5 by 28 and 29.

  • As non-organic chemists with deficiencies in nomenclature and numbering of compounds, we simply took the numbering the software ChemDraw provided (long time ago). A check in the literature helped us to find the correct way of numbering cholesteryl acetate (the one suggested by this referee).

P11L4 1JCH(112Hz for 1JC28H28  J value for terminal methyl groups should be larger than 120 Hz. Please check the values. Numbering should be also revised based on the revision of Figure 3.

  • Thank you for noting this typo. The correct value is 122 Hz. The numbering of both coupling constants has been accordingly adapted.

P12L11 methyl groups (~110Hz) => methyl groups (~125Hz)

  • Changed

P13L11 115Hz for 1JC8H8    Is it a real 1JCHvalue?  1JCHvalue for methyl group of toluene is 126Hz. Please check it.

  • Thank you for noting this typo. The correct value is 125 Hz.

Reviewer 2 Report

For the molecular structure elucidation and verification, 1D and 2D NMR spectra are used. In so doing it is necessary to carry out a reliable assignment of 13C NMR signals to resonances of Cq, CH, CH2 and CH3 groups. 2D NMR experiments (like HSQC and HMBC) are much more sensitive than 1D 13C spectral editing experiments such as APT, PENDANT, and DEPTQ, but 2D NMR spectra provide not sufficient resolution in the 13C indirect dimension. This potentially prevents the unambiguous differentiation between closely spaced 13C signals. The mentioned spectral edited 1D 13C experiments possess  higher resolution. As DEPTQ provide information about both Cq and protonated carbons it is chosen frequently to obtain the chemical shift and multiplicity information for all types of carbons if concentration of samples is enough. 
The authors of this work found that for molecules possessing large ranges of 1JCH coupling constants, the quality of DEPTQ135 spectra and edited subspectra was not satisfactory: many artifacts are observed from other CHn multiplicities and missing of very weak signals, especially CH3 or alkyne CH groups, can have place.
To overcome the mentioned drawbacks, the authors elaborated a new pulse sequence DEPTQ+ in which the evolution periods of length S = 0.5/1JCH are replaced by three spin echoes of different length, which allows one to tune experiment to different 1JCH values. This is especially valuable when a large range of 1JCH coupling constants is to be expected. 
 DEPTQ+ was   compared with the DEPTQ experiment, designed to unequivocally identify all carbon multiplicities in two experiments. The comparison confirmed advantages of   DEPTQ+. It was shown on examples that the DEPTQ+ pulse sequence provides unambiguous DEPTQ spectra with a leveling of all carbon intensities, which allows distinguish between real signals and weak signals of artifacts. The results are well grounded theoretically and corroborated experimentally. 
The limitations of this experiment are that it is effective when the molecules are of small or medium size and possess a wide range of 1JCH coupling constants, while the sample concentrations are high enough. The need for choosing the relaxation delay long enough to obtain reliable data is an additional limitation. Nevertheless, I suppose that the new experiment will be used especially for structure verification when the suggested molecular formula is known and the range of 1JCH couplings can be easily evaluated. 

Comments:

  1. Tables 1 and 2 have headers shown not in the due places.
  2. Figures 6 and 7 are presented to compare spectra of 4-methyl-N,N-di(prop-2-yn-1-yl)aniline obtained by DEPTQ and DEPTQ+ experiments. However, absence of structural formula and atom numbers on spectra hampers the comparison. I recommend to present Figures 6 and 7 like the Figure 1.
  3. P.2, repeated "that" should be deleted.

The manuscript is well written and thus publication in the Molecules is recommended with minor revisions.

Author Response

For the molecular structure elucidation and verification, 1D and 2D NMR spectra are used. In so doing it is necessary to carry out a reliable assignment of 13C NMR signals to resonances of Cq, CH, CH2 and CH3 groups. 2D NMR experiments (like HSQC and HMBC) are much more sensitive than 1D 13C spectral editing experiments such as APT, PENDANT, and DEPTQ, but 2D NMR spectra provide not sufficient resolution in the 13C indirect dimension. This potentially prevents the unambiguous differentiation between closely spaced 13C signals. The mentioned spectral edited 1D 13C experiments possess  higher resolution. As DEPTQ provide information about both Cq and protonated carbons it is chosen frequently to obtain the chemical shift and multiplicity information for all types of carbons if concentration of samples is enough. 
The authors of this work found that for molecules possessing large ranges of 1JCH coupling constants, the quality of DEPTQ135 spectra and edited subspectra was not satisfactory: many artifacts are observed from other CHn multiplicities and missing of very weak signals, especially CH3 or alkyne CH groups, can have place.
To overcome the mentioned drawbacks, the authors elaborated a new pulse sequence DEPTQ+ in which the evolution periods of length S = 0.5/1JCH are replaced by three spin echoes of different length, which allows one to tune experiment to different 1JCH values. This is especially valuable when a large range of 1JCH coupling constants is to be expected. 
 DEPTQ+ was   compared with the DEPTQ experiment, designed to unequivocally identify all carbon multiplicities in two experiments. The comparison confirmed advantages of   DEPTQ+. It was shown on examples that the DEPTQ+ pulse sequence provides unambiguous DEPTQ spectra with a leveling of all carbon intensities, which allows distinguish between real signals and weak signals of artifacts. The results are well grounded theoretically and corroborated experimentally. 
The limitations of this experiment are that it is effective when the molecules are of small or medium size and possess a wide range of 1JCH coupling constants, while the sample concentrations are high enough. The need for choosing the relaxation delay long enough to obtain reliable data is an additional limitation. Nevertheless, I suppose that the new experiment will be used especially for structure verification when the suggested molecular formula is known and the range of 1JCH couplings can be easily evaluated.

  • We thank this referee for his comments

Comments:

  1. Tables 1 and 2 have headers shown not in the due places.

  • We have moved both headers before the actual table.

  1. Figures 6 and 7 are presented to compare spectra of 4-methyl-N,N-di(prop-2-yn-1-yl)aniline obtained by DEPTQ and DEPTQ+ However, absence of structural formula and atom numbers on spectra hampers the comparison. I recommend to present Figures 6 and 7 like the Figure 1.

  • Indeed, we thank this reviewer for this useful advice. We have put the structure of 4-methyl-N,N-di(prop-2-yn-1-yl)aniline in figures 6 & 7 for clarity.

  1. 2, repeated "that" should be deleted.

  •  

The manuscript is well written and thus publication in the Molecules is recommended with minor revisions.

Reviewer 3 Report

The manuscript entitled "The DEPTQ+ Experiment: Leveling the DEPT Signal Intensities and Clean Spectral Editing for Determining CHn Multiplicities" by P. Bigler, C. Melendez-Becerra, and J. Furrer proposes a remedy to a frequently encountered problem, when one attempts to determine carbon multiplicities from DEPT-like spectra. The manuscript is well prepared and it is nowadays unexpected but refreshing to read an analytic pulse sequence analysis by Cartesian operator formalism calculations.

A few questions and remarks were noted during the reading of the manuscript, and should be considered for manuscript improvement.

Figure 1 (and other figures with spectra). Is there a technical reason to present low-resolution raster images? TopSpin can make nice vector drawings with the plot0 command. Please do not consider this remark if high resolution graphics were uploaded at manuscript submission time.

Figure 2. DETQ: The two central delta delays on the 13C channel are drawn as being not of equal duration. The corresponding Bruker pulse sequence make the 180(1H) and 90(13C) pulses in the middle of the sequence consecutive but not centered, as drawn in your Fig. 2. Could these pulses be simultaneously applied by symmetry? Is the first "decoupling" really a decoupling or a nOe build-up sequence? Is it applied at the same power level as the decoupling applied during signal acquisition? At the very end of the figure caption, "(DEPTQ4590)" should be "(DEPTQ+4590)".

page 5, equation 5: Cx appears here for the first time but is never defined.

page 9, table 1, line 4. The Cx factors are missing.

page 10, figure 3. What is the point in numbering the carbon atoms, moreover with this unusual numbering scheme?

page 15, practical aspects. In the perspective of running the DEPTQ+ pulse sequence in an automated environment with unsupervised sample changer operation, what is the influence of pulse mis-calibration on the quality of multiplicity editing, especially when the probehead is not systematically re-tuned and re-matched on the 1H and 13C channels? This question may go beyond the scope of the manuscript, so no complementary experiment is requested. However, a sentence related to this topic would be appreciated. In the rewiever's hands, the DETP sequence produces spectra that are somewhat dependent on pulse length proper calibration.

page 15, synthesis. The description of the NMR spectra does not report how they were referenced. Was this description made by yourself or borrowed from another source? Immediately after, you write that all experiments were recorded on a 500 MHz (1H) NMR spectrometer.

page 15, NMR measurements. 128k is probably the number of real data points. I guess TD = SI = 128k and that you use one level of zero-filling. Sorry for writing that, but "spectral width" would be better than "sweep width" because there is nothing sweeping there.

Figure S2, why is the x-axis limited to 150 Hz when the optimal 1J is 185 Hz? Please consider this question for other Intensity(J) graphs in the SI document.

Author Response

Ref 3

The manuscript entitled "The DEPTQ+ Experiment: Leveling the DEPT Signal Intensities and Clean Spectral Editing for Determining CHn Multiplicities" by P. Bigler, C. Melendez-Becerra, and J. Furrer proposes a remedy to a frequently encountered problem, when one attempts to determine carbon multiplicities from DEPT-like spectra. The manuscript is well prepared and it is nowadays unexpected but refreshing to read an analytic pulse sequence analysis by Cartesian operator formalism calculations.

A few questions and remarks were noted during the reading of the manuscript, and should be considered for manuscript improvement.

Figure 1 (and other figures with spectra). Is there a technical reason to present low-resolution raster images? TopSpin can make nice vector drawings with the plot0 command. Please do not consider this remark if high resolution graphics were uploaded at manuscript submission time.

  • We have to check with the editing team. So far, the presented figures have been prepared with TopSpin and exported as tiff images with a 600 dpi resolution.

Figure 2. DETQ: The two central delta delays on the 13C channel are drawn as being not of equal duration. The corresponding Bruker pulse sequence make the 180(1H) and 90(13C) pulses in the middle of the sequence consecutive but not centered, as drawn in your Fig. 2. Could these pulses be simultaneously applied by symmetry?

  • This is a very good question, which has been addressed in the original paper ( Reson. Chem. 2007, 45, 469-472): the best results are indeed obtained when the 180(1H) and 90(13C) pulses are applied consecutively and not centred. Note that this is not the case for the DEPTQ+: As for instance in INEPT, the the 180(1H) and 180(13C) pulses can be applied simultaneously (Centred).

Is the first "decoupling" really a decoupling or a nOe build-up sequence? Is it applied at the same power level as the decoupling applied during signal acquisition?

  • It is a nOe build-up period. The power level applied is the same than that used during acquisition.

At the very end of the figure caption, "(DEPTQ4590)" should be "(DEPTQ+4590)".

  • Thank you, corrected.

page 5, equation 5: Cx appears here for the first time but is never defined.

  • We have added an explanation in the sentence preceding equation 5.

page 9, table 1, line 4. The Cx factors are missing.

  • Added

page 10, figure 3. What is the point in numbering the carbon atoms, moreover with this unusual numbering scheme?

  • 2 coupling constants are mentioned in the text subsequently to this table. The numbering (which is now correct) was modified according to the comments of Ref 1 (see above).

page 15, practical aspects. In the perspective of running the DEPTQ+ pulse sequence in an automated environment with unsupervised sample changer operation, what is the influence of pulse mis-calibration on the quality of multiplicity editing, especially when the probehead is not systematically re-tuned and re-matched on the 1H and 13C channels? This question may go beyond the scope of the manuscript, so no complementary experiment is requested. However, a sentence related to this topic would be appreciated. In the rewiever's hands, the DETP sequence produces spectra that are somewhat dependent on pulse length proper calibration.

  • Thank you for this remark and useful advice. Indeed, DEPT sequences produce spectra that are dependent on pulse length proper calibration. However, as shown in our recent review ( Rep. NMR Spectrosc., 2017; 92, 1-82.), DEPT are less sensitive to poorly calibrated 1H and 13C pulses. Compared to other editing sequences (INEPT, PENDANT, SEMUT, APT). Note also that the critical 13C 180 degree pulses are nowadays automatically replaced by an adiabatic refocusing pulse to improve the refocusing effectiveness over a wide bandwidth. This enhances particularly the intensities and phases of signals close to the spectral window’s edges.

We have added this sentence in the text.

page 15, synthesis. The description of the NMR spectra does not report how they were referenced.

  • Classical reference with respect to TMS. Added.

Was this description made by yourself or borrowed from another source?

  • It’s our own analysis, that match the spectra shown in figures 1, 6 and 7.

Immediately after, you write that all experiments were recorded on a 500 MHz (1H) NMR spectrometer.

  • Indeed, we change All to All DEPTQ and DEPTQ+ experiments

page 15, NMR measurements. 128k is probably the number of real data points. I guess TD = SI = 128k and that you use one level of zero-filling. Sorry for writing that, but "spectral width" would be better than "sweep width" because there is nothing sweeping there.

  • But for the Chirp pulses, “sweep” is correct.

Figure S2, why is the x-axis limited to 150 Hz when the optimal 1J is 185 Hz? Please consider this question for other Intensity(J) graphs in the SI document.

  • Figure S2 shows the specific behaviour of CH3 groups, for which the 1J barely exceeds 135 Hz and reach 150 Hz in very few cases. For this reason, the x-axis has been on purpose limited to 150 Hz, even if the optimization is done with 185 Hz. The same holds true for the other graphs in the SI.